# Transfer learning for atomistic simulations using GNNs and kernel mean embeddings

**John I. Falk**
CSML
Istituto Italiano di Tecnologia
Genova, Italy
me@isakfalk.com

**Luigi Bonati**
Atomistic Simulations
Istituto Italiano di Tecnologia
Genova, Italy
luigi.bonati@iit.it

**Pietro Novelli**
CSML
Istituto Italiano di Tecnologia
Genova, Italy
pietro.novelli@iit.it

**Michele Parrinello**
Atomistic Simulations
Istituto Italiano di Tecnologia
Genova, Italy
michele.parrinello@iit.it

**Massimiliano Pontil**
CSML
Istituto Italiano di Tecnologia
Genova, Italy
University College London, U.K.
massimiliano.pontil@iit.it

## Abstract

Interatomic potentials learned using machine learning methods have been successfully applied to atomistic simulations. However, accurate models require large training datasets, while generating reference calculations is computationally demanding. To bypass this difficulty, we propose a transfer learning algorithm that leverages the ability of graph neural networks (GNNs) to represent chemical environments together with kernel mean embeddings. We extract a feature map from GNNs pre-trained on the OC20 dataset and use it to learn the potential energy surface from system-specific datasets of catalytic processes. Our method is further enhanced by incorporating into the kernel the chemical species information, resulting in improved performance and interpretability. We test our approach on a series of realistic datasets of increasing complexity, showing excellent generalization and transferability performance, and improving on methods that rely on GNNs or ridge regression alone, as well as similar fine-tuning approaches.

## 1 Introduction

Atomistic simulations have become a pillar of modern science and are pervasively used in many areas of physics, chemistry, and biology. Among these techniques, molecular dynamics plays a prominent role. This method simulates the time evolution of a system of atoms by integrating Newton's equation of motion [1]. The forces acting on the particles are determined from a model for the interactions, called the potential energy surface (PES), on whose accuracy the reliability of the simulation depends. For a long time, the interactions were modeled in a rather empirical, and of course, not very accurate way [1, 2]. A significant step forward was made with the introduction of *ab initio* molecular dynamics in which the interactions are computed on the fly from accurate electronic structure calculations [3, 4]. This implies solving at every step the Schroedinger equation, typically with the use of some approximation such as the popular Density Functional Theory (DFT) scheme [5]. This approach is much more accurate but at the same time more computationally expensive, limiting the system size (i.e. the number of atoms) and the time scale that can be simulated.

Starting with the work of Behler and Parrinello [6], machine learning potentials have emerged as promising candidates to alleviate the tension between accuracy and efficiency [7, 8]. They regress the

37th Conference on Neural Information Processing Systems (NeurIPS 2023).

potential energy, as a function of the atomic positions and the chemical species, on a (large) set of expensive *ab initio* calculations. Once successful, this strategy results in an *ab initio*-quality model of the potential energy at a fraction of the cost, speeding up simulations by orders of magnitude.

For this procedure to be successful, however, a good representation of the physical system, including its symmetries, is necessary. The use of handcrafted physical descriptors [6, 9–11] is often a laborious procedure that limits applicability to systems with few chemical species. In recent years, graph neural networks (GNNs) have proven to be a viable alternative for directly representing the chemical environment, capable of scaling to large datasets with numerous chemical species and encoding symmetries directly in the architecture, e.g., through SE(3)-equivariant layers [12–16]. Nevertheless, obtaining an accurate model for real-life applications is as of today still a challenging task, requiring high-quality data samples which are scarce and/or very expensive to obtain.

As shown by the recent advancements in large language modeling [17, 18] and, before that, image classification [19, 20], fine-tuning a pre-trained large-scale representation of the data provides a highly effective paradigm to solve downstream tasks for which only a handful of data points are available. This approach paved the way for the concept of *foundation models* [21], at the core of the current generative AI revolution. Mirroring these developments, in this work we leverage the representation power of GNNs trained on large datasets of molecular configurations. In particular, we rely on the Open Catalyst dataset [22], which contains DFT relaxations for $\sim$1.2M catalytic systems, totaling over 260M data points. We show how, by exploiting the availability of this heterogeneous dataset, it is possible to learn interatomic potentials for specific systems taken from realistic chemical applications in a fast and data-efficient manner.

**Contributions** This paper makes the following contributions:

- We propose a transfer learning algorithm, which we refer to as mean embedding kernel ridge regression (MEKRR), for modeling the potential energy surface of atomic systems. MEKRR combines GNN representations pre-trained on large datasets with fine-tuning via kernel mean embeddings. This combination allows to satisfy the physical symmetries inherent to atomistic systems. Specifically, GNN features take care of roto-translational invariance, while kernel mean embeddings are chosen to satisfy the permutational symmetry.

- We introduce a new kernel function in the context of modeling potential energy surfaces, which exploits chemical species information. This shows superior performance and facilitates monitoring the chemical evolution of the system.

- We demonstrate excellent transferability and generalization performance on increasingly complex datasets. Remarkably, they include configurations sampled out-of-distribution with respect to the GNN pre-trained representation.

**Related work** There is a long list of relevant works (see also [7] and references therein) on representing the potential energy surface by machine learning methods. In particular, the first models employed a set of physical descriptors in combination with either neural networks [6] or kernel methods [23–25]. Later, it was proposed to model with neural networks the descriptors as well [26]. Recently, graph networks have been used to directly represent physical systems and regress energy and forces [12, 15, 27, 28]. A combination of GNN features and kernel-based methods has been investigated in [29], but without doing transfer learning from a larger dataset.

In terms of transfer learning, there is a long line of work in transferring deep representations on images, see [19, 20] and references therein. Current progress in few-shot image classification, where algorithms can adapt quickly to new classification problems, can partially be attributed to adding a preprocessing step where each image is mapped through a meta-learned or pre-trained representation, see e.g. [30–36]. In particular, [36] employs kernel ridge regression (KRR) on top of a meta-learned feature map. More recently, the same ideas have been applied to language modeling (see e.g. [17, 18]), revolutionizing the field. Pre-training strategies have started very recently to emerge also in the context of interatomic potentials, as a way to interpolate across different levels of theory [37–39] or to exploit the release of large and heterogeneous datasets such as OC20 [40].

**Organization** In Section 2 we specify the machine learning problem we are aiming to solve. In Section 3 we introduce the kernel mean embedding framework together with KRR and in Section 4 we introduce our method MEKRR. In Section 5 we validate our method on a variety of realistic datasets of increasing complexity. Finally, in Section 6 we conclude and outline future directions.

## 2 Learning the potential energy surface from atomistic simulations

### 2.1 Setting

We consider systems composed of $n$ atoms and $S$ different chemical species, described via one-hot-encoding over $[S]$, where $[S] = \{1, \ldots, S\}$. Each atom is described by its Euclidean position $r \in \mathbb{R}^3$ and its chemical species $z \in \{0, 1\}^S$. We denote a state of the system by $x = (r_i, z_i)_{i=1}^n = (R, Z) \in \mathcal{X}$ using the design matrices $R \in \mathbb{R}^{n \times 3}$ and $Z \in \{0, 1\}^{n \times S}$.

The quantity we want to regress is the potential energy, which is a scalar function $E : \mathcal{X} \to \mathbb{R}$. The target values $E_t$ are calculated by querying the *ab initio* method of choice, DFT in the case of the OC20 dataset. A training dataset is therefore composed by a sequence of $T$ atomic configurations $(x_t)_{t=1}^T$ together with the corresponding labels $(E_t)_{t=1}^T$.

The physics of the system posits that any estimator of the energy $\hat{f} : \mathcal{X} \to \mathbb{R}$ should possess certain invariant properties reflecting the underlying physical symmetries. These invariances are *roto-translational invariance* with respect to the position vectors and *permutation-invariance within each group of chemical species*. In addition, the energy estimator should be curl-free and smooth [27]. A wealth of previous works introduced deep learning architectures incorporating these invariances by design [e.g. 16, 27, 41, 42]. Our transfer-learning scheme works by exploiting these invariant architectures so that the final fine-tuned estimator is invariant as well.

### 2.2 Objective function

In order to learn potential energies, one considers models $f_w$ from a set of functions (e.g., a reproducing kernel Hilbert space or neural network functions) parameterized by $w$ and fitted to a given dataset $(x_t, E_t)_{t=1}^T$. This is achieved by minimizing an objective function, which can be split into a data-fitting term and a regularizer which encourages functions of low complexity.

Typically, the data fitting term is a least squares loss between outputs and predictions. In this case the regularized empirical risk minimization reads

$$\hat{\mathcal{R}}(w) = \sum_{t=1}^T \left( (1-\gamma)(E_t - f_w(R_t, Z_t))^2 + \gamma \| F_t + \nabla_{R_t} f_w(R_t, Z_t) \|^2 \right) + \lambda \Omega(w), \qquad (1)$$

for some regularizer $\Omega$ and regularization parameter $\lambda > 0$ and a loss weight $\gamma \in [0, 1]$. A loss weight $\gamma \neq 0$ is used whenever the forces $F(x, r) = -\nabla_r E(x)$ are regressed alongside the energy. Typically we will use $\Omega(w) = \|w\|^2$ as the regularizer, but in Section 4.3 we extend this to incorporate additional information. Since in the experiments discussed below we focus on energy prediction, we let $\gamma = 0$.

## 3 Kernel ridge regression and kernel mean embeddings

Our method is designed around kernel methods, a well-established tool at the heart of most non-parametric machine learning algorithms [43, 44]. A kernel $K$ is any positive definite function (see 44, Definition 4.15) $K : \mathcal{X} \times \mathcal{X} \to \mathbb{R}$ on the input space $\mathcal{X}$. For any kernel it exists a *feature space* $\mathcal{H}_K$ and a *feature map* $\phi : \mathcal{X} \to \mathcal{H}_K$ such that $K(x, x') = \langle \phi(x), \phi(x') \rangle$ for all $x, x' \in \mathcal{X}$.

**Kernel ridge regression** is a supervised learning algorithm that parametrize the dependence between inputs $x \in \mathcal{X}$ and scalar outputs $y \in \mathbb{R}$ as $y = \langle w, \phi(x) \rangle$ for some vector $w$ in the feature space. Given a dataset $(x_t)_{t=1}^T \in \mathcal{X}^T$ of $T$ input points, and a corresponding dataset of outputs $(y_t)_{t=1}^T \in \mathbb{R}^T$, KRR learns a functional relation between inputs and outputs by solving the least squares minimization problem

$$\hat{w} = \operatorname*{argmin}_{w \in \mathcal{H}_k} \left\{ \sum_{t=1}^T (\langle w, \phi(x_t) \rangle - y_t)^2 + \lambda \|w\|_{\mathcal{H}_K}^2 \right\}. \qquad (2)$$

A basic result known as the *representer theorem* (see e.g. 44, Theorem 5.5) prescribe that the solution of (2) is just a linear combination of the feature map evaluated at the training points, that is $\hat{w} = \sum_{t=1}^T c_t \phi(x_t)$ for some $(c_t)_{t=1}^T \in \mathbb{R}^T$. The representer theorem and the fact that $K(x, x') = \langle \phi(x), \phi(x') \rangle$ also imply that predicting any new point $x$ with KRR is as simple as evaluating

$\langle \hat{w}, \phi(x) \rangle = \sum_{t=1}^{T} c_t \langle \phi(x_t), \phi(x) \rangle = \sum_{t=1}^{T} c_t K(x_t, x)$. KRR can also be readily extended [45] to regress the gradient of $y$ with respect to $x$, a technique used e.g. in [29] to concurrently learn potential energies and forces.

In the standard case in which the input space is a subset of an Euclidean space $\mathcal{X} \subseteq \mathbb{R}^d$, many kernels have been designed; see, for instance, 46, Section 4.2. This notwithstanding, kernel functions can be defined on arbitrary input spaces $\mathcal{X}$. In the case of atomic configurations, the potential energy is invariant under the permutation of atoms with the same chemical species. It is therefore advantageous to design kernels able to preserve such symmetry. We now introduce the concept of kernel mean embeddings [47], which allow us to define permutationally-invariant kernels over atomic configurations.

**Kernel mean embeddings** consider the case in which inputs $x$ are sets of points living in an Euclidean space $x = \{r_i \in \mathbb{R}^d \colon i \le n\}$. For example, an atomic configuration is a set containing the position of each atom in the system ($d = 3$). For any feature map $\psi(r)$ and the corresponding kernel function $k(r, r') = \langle \psi(r), \psi(r') \rangle$ on points $r \in \mathbb{R}^d$ we can define a feature map acting on the whole set $x$ as

$$\phi(x) := C_x \sum_{r_i \in x} \psi(r_i)$$

with $C_x > 0$ a positive normalization constant. We note in passing that the value of $C_x$ depends on the chemical property that we want to learn. Indeed, setting $C_x = 1$ returns extensive properties such as the potential energy, while $C_x = 1/n$ is appropriate to model intensive ones. This allows us to define kernels on sets $x$ as

$$K(x, x') = \langle \phi(x), \phi(x') \rangle = C_x C_{x'} \sum_{\substack{r_i \in x, \\ r_j' \in x'}} \langle \psi(r_i), \psi(r_j') \rangle = C_x C_{x'} \sum_{\substack{r_i \in x, \\ r_j' \in x'}} k(r_i, r_j'). \tag{3}$$

Using the definition of kernel mean embeddings (3) inside the KRR algorithm (2) enables us to learn scalar functions over sets of points. In the following section, we will combine these algorithms together with pre-trained feature maps to define a principled and efficient method to learn potential energy surfaces from data.

## 4    Method

Our method relies on pre-trained GNN representations of chemical environments and uses them as feature maps to define a kernel mean embedding (3) acting on atomic configurations. In practice, we do so by leveraging pre-trained feature maps on the OC20 dataset [22]. Once a kernel based on mean embeddings is defined, we can use KRR to regress the potential energy from sampled data (see Fig. 1 for a diagram of the method). In this respect, we note that our method shares similarities with the idea behind foundation models [21], where a representation trained on large datasets is transferred to novel settings by means of fine-tuning. While in this work we show the performance of pre-trained representations based on SCN [14] and SchNet [27], any other GNN architecture is a perfectly valid choice, and can be used in place of ours without the need of further adjustments.

### 4.1    GNN representations of chemical environments

SCN and SchNet are instances of graph neural networks (GNNs) [48–50], a class of architectures designed to learn mappings over graphs. A graph $\mathcal{G} = (\mathcal{V}, \mathcal{E})$ is a collection of $n$ nodes $\mathcal{V}$ and edges $\mathcal{E} \subseteq \mathcal{V} \times \mathcal{V}$ between pairs of nodes. Representing the chemical environment with GNNs involves a preprocessing step that turns a configuration $x$ into a graph by associating each atom to a node and constructing the edges according to the matrix of pairwise distances between atoms. Each node $i$ is then initialized to a feature vector $h_i$ encoding the chemical species of the atom via a (possibly learnable) embedding layer. Each GNN layer then updates the node features via a nonlinear message-passing scheme

$$h_i \mapsto g_\theta(h_i, \sum_{j \in \mathcal{N}(i)} \eta_\theta(h_i, h_j)).$$

Here, $\mathcal{N}(i)$ is the neighborhood of $i$ i.e. the set of nodes connected to $i$ through an edge, $g_\theta$ is a learnable vector-valued function such as an MLP and $\eta_\theta$ is a message-passing function. A graph

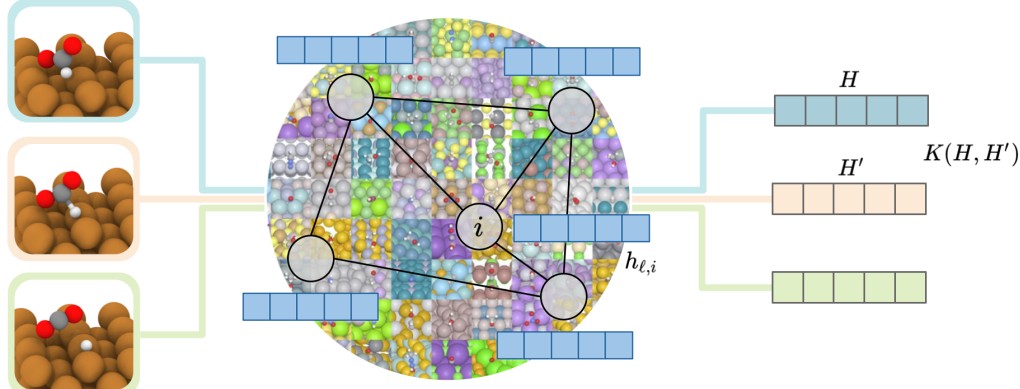

a. Atomic feature maps from a pretrained GNN    b. Transfer learning via KRR

Figure 1: Diagram of MEKRR.

neural network is formed by concatenating multiple message-passing layers. The weights of the embedding and GNN layers are then learned in an end-to-end fashion typically using first-order optimization.

Among the many GNN architectures available [51], a growing number is being developed to specialize in computational chemistry settings, see for instance [16, 28, 41, 42, 52–56].

## 4.2   Our method: mean embeddings of GNN representations

We are now ready to combine together the techniques described so far and describe in full detail our method. Starting with an atomic configuration $x$, we pass it to a pre-trained GNN representation to get a collection of node features

$$x \mapsto H(x) := (h_i(x))_{i=1}^n.$$

In the case of equivariant GNNs we additionally need to extract invariant features from equivariant ones via a pooling method such as averaging. In case of SCN [14], for example, we average the features over the sphere (see Appendix C.2). We then evaluate the kernel mean embedding of the node features

$$H(x) \mapsto \phi(H(x)) = C_x \sum_{h_i(x) \in H(x)} \psi(h_i(x)).$$

Finally, at training time we can use the kernel mean embedding $\phi(H(x))$ to solve the KRR problem as

$$\hat{w} = \underset{w \in \mathcal{H}_k}{\operatorname{argmin}} \left\{ \sum_{t=1}^T (\langle w, \phi(H(x_t)) \rangle - E_t)^2 + \lambda \|w\|_{\mathcal{H}_K}^2 \right\}, \tag{4}$$

while at inference time we can get an estimation for the potential energy at $x$ by simply evaluating $\langle \hat{w}, \phi(H(x)) \rangle$.

Some practical considerations are now in order. Given a pre-trained GNN representation, one is not bound to use the node features of the last layer. Intermediate representations have the benefit of being faster to evaluate, and we empirically observed that the performance of MEKRR is not hindered when truncating the GNNs pre-trained on OC20 at an intermediate layer (see Appendix C). Furthermore, there are many strategies to solve the KRR problem (4) based e.g. on the solution of a linear system or on gradient-descent. In Algorithm 1 we report the pseudo-code describing our implementation of the training and prediction steps of MEKRR.

## 4.3   Chemically-informed kernel mean embeddings

The chemical species of each atom is implicitly encoded in the GNN representation through its initial layer, embedding each atom of the same chemical species to the same dense feature vector. In MEKRR we strengthen this dependence by encoding the same information again at the level of mean embeddings. For a configuration $x$ let now $H_s(x) := \{h_i(x) \mid \text{species of } i = s\}$ be the subset

---

**Algorithm 1** Mean Embedding Kernel Ridge Regression (MEKRR)

---

**Parameters**: Training dataset $(x_t, E_t)_{t=1}^T$ of configurations $x_t$ and potential energies $E_t$. Kernel function $k$. Interpolation parameter of the multi-weight formulation $\alpha \in [0, 1]$. Tikhonov regularization $\lambda > 0$. Pre-trained GNN feature map $x \mapsto H(x)$.

**function** TRAIN
    $(H(x_t))_{t=1}^T \leftarrow (x_t)_{t=1}^T$                                 ▷ Evaluate input representations
    $G_{t,l} \leftarrow K_\alpha(H(x_t), H(x_l))$             ▷ Form the kernel matrix using (3) and/or (5)
    $c \leftarrow (G + \lambda I)^{-1} E$         ▷ Compute KRR coefficients where $E = (E_1, \ldots, E_T)^\top$
    **return** $c$
**end function**

**function** PREDICT$(x)$
    $H(x) \leftarrow x$                                     ▷ Evaluate GNN representation
    $v(x) \leftarrow (K_\alpha(H(x), H(x_t)))_{t=1}^T$                   ▷ Form the kernel matrix
    **return** $c^\top v(x)$                            ▷ Return the KRR prediction
**end function**

---

of node features corresponding to atoms of the $s$'th chemical species. We then consider a composite kernel

$$K_\alpha(H(x), H(x')) := (1 - \alpha)K\big(H(x), H(x')\big) + \alpha \sum_{s=1}^S K\big(H_s(x), H_s(x')\big). \tag{5}$$

The kernel $K_\alpha$ is a direct generalization of the mean embedding kernel (3) where the parameter $\alpha$ allows to interpolate between emphasizing all atomic interactions equally ($\alpha = 0$) and only within-species atomic interactions ($\alpha = 1$).

Solving KRR (4) with the composite kernel (5) is also equivalent to considering an extension of (4) where to each chemical species is given its own weight while encouraging the weights to be close to each other and of small magnitude through a regularizer. Precisely, let $(w_s)_{s=1}^S$ be the set of displacements from a center weight $w_0$ for each chemical species. The weight of a chemical species $s$ is given by $w_s + w_0$ and the potential energy function for a configuration $x$ takes the form $\sum_{s=1}^S \langle w_s + w_0, \phi(H_s(x) \rangle$. In Appendix A we provide a full characterization of this extension, highlighting its connection to multi-task learning [57, 58], in which multiple tasks are learned jointly.

## 5 Experiments

In this section, we consider the realistic problem of modeling the potential energy surface of catalytic reactions occurring on metallic surfaces. We evaluate our approach against methods that are representative of the KRR and GNN approaches, testing them on datasets of increasing complexity taken from realistic applications. As is customary in the literature, we employ the root mean squared error (RMSE) normalized by the number of atoms as a metric, to facilitate the comparison between systems of different sizes. We make the code repository available at `https://github.com/IsakFalk/atomistic_transfer_mekrr`.

### 5.1 Baselines and MEKRR

We consider baselines spanning different categories. Firstly we examine supervised learning algorithms trained from scratch on the provided datasets, either through GNNs or kernel methods with hand-crafted physical features. In this category we have Schnet, SCN and GAP. **SchNet** [59] is one of the first GNNs to be applied successfully to chemistry, which uses a radial basis function representation of the interatomic distances. Spherical Channel Networks (**SCN**) [14] is another GNN and its atom embeddings are a set of spherical functions represented via spherical harmonics. SCN is one of the state-of-the-art models on the OC20 dataset. For both SchNet and SCN we use the codebase of [22]. Finally, Gaussian Approximation Potential (**GAP**) is a kernel-based method that builds a Gaussian Process using the Smooth Overlap of Atomic Positions (SOAP) descriptors [60], which we use through the QUIP/quippy code base [61, 62].

The second category of baselines concerns transfer learning methods based on the OC20 dataset [22]. In this case we consider the fine-tuning of Schnet (**Schnet-FT**) which is done by keeping the parameters fixed up to the representation used for MEKRR and then optimizing the subsequent layers on the new dataset.

These baselines are tested against our method (**MEKRR**), which uses a kernel mean embedding with Gaussian kernel based on different pre-trained GNN features. The length-scale of the Gaussian kernel is chosen according to the median heuristic [63]. We will denote MEKRR-(SchNet) and MEKRR-(SCN) the variants using Schnet and SCN node features as inputs, respectively.

## 5.2 Datasets

We first describe the dataset which has been used for the construction of the pre-trained GNN feature map (**OC20**), and then present the system-specific MD datasets where our method is fine-tuned on (**Cu/formate**, **Fe/N$_2$**).

**OC20** The Open Catalyst (OC) 20 is a large dataset of *ab initio* calculations aimed at estimating adsorption energies on catalytic surfaces. It comprises ~250 millions of DFT calculations, generated from over 1.2 million relaxations trajectories of different combinations of molecules and surfaces. In each relaxation, the positions of the molecule and of the surface upper layers are optimized via gradient descent in order to compute the adsorption energy. The adsorbate is selected out of 82 molecules relevant to environmental applications, while, for each of them, up to $55^3$ surfaces are selected, including binary and ternary compounds. We underline that, for each adsorbate-surface pair, the configurational space sampled is very limited, and especially it does not cover out-of-equilibrium and reactive (e.g. bond forming or breaking) events.

We fine-tune the method and test it on two datasets that are representative of reactive catalytic events, obtained by means of molecular dynamics simulations coupled with enhanced sampling methods [64, 65] to avoid mode collapse into the local minima of the potential energy landscape. Indeed, whereas the OC20 dataset contains short, correlated relaxations toward the nearest equilibrium state, typical catalytic reaction datasets require sampling all local minima (adsorption states of the molecule) and especially reactive events, in which, due to interaction with the surface, bonds can be broken or formed. For this reason, these applications are challenging as they relate to realistic datasets containing mostly reactive events that are outside the distribution of the OC20 dataset. We split all the below datasets into a train, validation, and test set using random splitting of $60/20/20$.

**Cu/formate** The first dataset is a collection of molecular dynamics simulations of the dehydrogenation reaction of formate on a copper (Cu) <110> surface [15], initialized along the reaction path (obtained with the Nudged Elastic Band method [65]), in which the molecule loses its hydrogen atom upon interaction with the surface.

**Fe/N$_2$ ($D_i$)** The second dataset consists of molecular dynamics simulations of a nitrogen molecule adsorbing on an iron (Fe) <111> surface at high temperature (T = 700 K) and breaking in two nitrogen atoms [66]. A peculiarity of this dataset is that it contains data from different sources (e.g. standard and biased molecular dynamics) and system sizes, allowing us to also assess the transferability of the methods across different conditions. For this reason, we divide it into 4 subsets, denoted with $D_i$:

$D_1$ : **AI-MD** *Ab initio* molecular dynamics simulations. The resulting configurations are highly correlated and cover a small portion of the configurational space related to the adsorption process, thus being the closest dataset to the OC20 one.

$D_2$ : **AI-METAD** Here the *ab initio* MD simulation is accelerated with the help of the metadynamics [64] technique. This is an importance sampling method that allows rare events to be observed, and thus it has been employed for collecting reactive configurations in the training set [67]. Due to the metadynamics approach, a larger region of configurational space is sampled with respect to $D_1$, allowing one to sample one bond-breaking event.

$D_3$ : **AL-METAD** Dataset built from an active learning procedure using an ensemble of NNs combined with metadynamics. In this simulation, multiple reaction events are observed, covering a wider region of the configurational space and providing a large number of

uncorrelated samples. Hence, these configurations are far from those used to pre-train the feature map.

$D_4$ : **AL-METAD-72** Same as $D_3$ but the surface is composed of 72 atoms (8 layers) to test the transferability across systems of different sizes.

## 5.3 Interpolating between shared and independent weights

The $\alpha$ parameter in the $K_\alpha$ kernel can vary in the range $[0, 1]$, which are the limiting cases between a shared or independent set of weights for each chemical species. We use cross-validation to set this parameter in practice. To initially fit the regularization parameter $\lambda$ we set $\alpha = 0$ and cross-validate $\lambda \in \{10^{-3}, \ldots, 10^{-9}\}$ using the same datasets. Despite this simple heuristic cross-validation scheme, as we will see, the scheme is effective, which we believe is a strength as it shows that the MEKRR method is simple to tune while still having the strongest performance among the competitors.

In Fig. 2 we show the cross-validation curves for MEKRR-(SchNet), related to the two datasets Cu/formate and Fe/N$_2$. In the latter case, we perform the cross-validation only on ($D_2$), which is representative of the family of datasets, and then use the found parameters also for the other datasets. The two plots show different behavior with the optimal $\alpha$ for the Cu/formate dataset occurring around $10^{-2}$. This means that the potential energy can be well described with shared weights across chemical species together with a small perturbation. Instead, in the Fe/N$_2$ dataset the optimal $\alpha$ occurs at the boundary leading to a kernel in which the weights for the two species are learned independently.

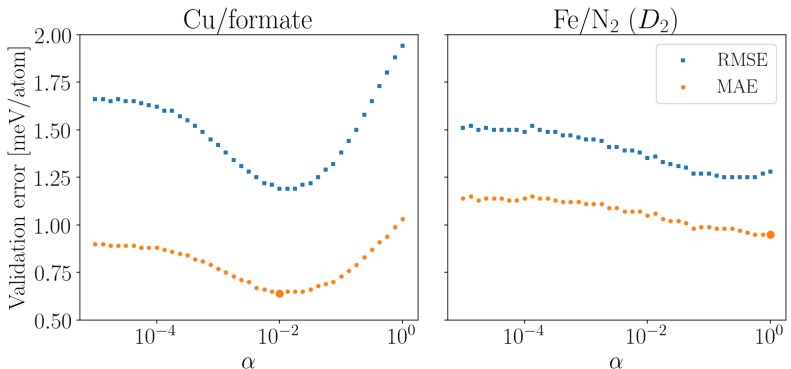

Figure 2: Validation error (RMSE / MAE) of MEKRR-(SchNet) on the Cu/formate and Fe/N$_2$ ($D_2$) datasets as a function of $\alpha$ geometrically spaced on a grid from 0 to 1 with optimal $\alpha$ and error given by a bold orange point. The optimal $\alpha$ for the Cu/formate dataset is positive but close to zero while the optimal $\alpha$ for the Fe/N$_2$ is found at the boundary at 1.0 leading to a hard multi-embedding kernel. We see that tuning the $\alpha$ allows for improved performance in practice and that the multi-weight formulation (5) is practically beneficial.

## 5.4 Potential energy regression

In this section, we consider the setting of predicting the potential energy surface. We first evaluate the performance of the models in predicting the energy for each of the datasets and then assess their generalization performance through the transfer learning setting, where we train and test on similar but distinct datasets.

**Same-dataset energy prediction** From Table 1 we see that MEKRR achieves the best performance in all datasets, both when the input features are extracted from SchNet and SCN. We note that, in general, the transfer learning algorithms (SchNet-FT and the two MEKRR variants) outperform the ones trained from scratch, with MEKRR being significantly faster (see Appendix B). Furthermore, it is worth highlighting that our method performs better than the baselines even when it is applied to datasets that are out-of-distribution for the pre-trained feature map. This is particularly evident for $D_3$ and $D_4$ which contain multiple reactive events (bond-breaking) that were never seen in the relaxations composing the OC20 dataset. This demonstrates the ability of the GNN trained on large and heterogeneous datasets to effectively represent chemical environments.

Table 1: Same-dataset energy prediction, metric being RMSE. The errors are in units of meV/atom. Best performance given by **bold** number in gray cell.

| Group | Algorithm | Fe/$N_2$ | | | | Cu/formate |
|-------|-----------|-------|-------|-------|-------|------------|
| | | $D_1$ | $D_2$ | $D_3$ | $D_4$ | |
| Supervised | GAP | 0.4 | 2.1 | 3.9 | 4.9 | 2.8 |
| | SchNet | 0.5 | 4.1 | 5.1 | 6.2 | 6.0 |
| | SCN | 0.3 | 5.1 | 7.5 | 7.3 | 2.5 |
| Fine-tune | SchNet-FT | **0.1** | 2.0 | 2.5 | 3.2 | 1.9 |
| Ours | MEKRR-(SchNet) | **0.1** | 1.3 | 2.4 | 3.3 | **1.2** |
| | MEKRR-(SCN) | 0.2 | **0.9** | **1.9** | **2.7** | 1.7 |

**Across-dataset energy prediction** Here we evaluate the performance of the algorithms and MEKRR on transferring from different systems in the Fe/$N_2$ family of datasets. To do this we consider the task of zero-shot transfer learning (see e.g. [18] and references therein) where we evaluate a model trained on a source dataset $D_{\text{source}}$ on a target dataset $D_{\text{target}}$. While the two datasets $D_{\text{source}}$ and $D_{\text{target}}$ may be sampled from arbitrary systems, we consider here systems that share some characteristics as we are evaluating the transfer capability of the models [68, 69]. Due to the ordering of the datasets $D_1, \ldots, D_4$ in increasing complexity on several axes (size, *ab initio* vs. active sampling, standard vs. biased dynamics, etc.) we consider a transfer from simpler to more complicated systems. Successfully transferring from simpler to more complicated systems has real-world impact as it can alleviate the high computational cost required for labeling via DFT calculations by reducing the number of points. From Table 2 we see that MEKRR-(SCN) has the lowest error in four out of five tasks, while MEKRR-(SchNet) has the lowest error in the remaining task. Furthermore, the relative transferability of MEKRR compared to the other methods even improves as the task becomes harder. To this respect, it is worth noting that $D_1, D_2, D_3$ are qualitatively similar, being all composed of 5 layers of Fe and differing for the sampling method used. Instead, the atomic environments contained in $D_4$ are different as they refer to a slab with a different number of layers. This explains the different order of magnitudes in the last two columns. Despite this, MEKRR still performs very well compared to the baselines.

Table 2: Transfer evaluation of algorithms on source to target: $D_{\text{source}} \rightarrow D_{\text{target}}$, metric being RMSE. The errors are in units of meV/atom. Best performance is given by **bold** number in gray cell.

| Group | Algorithm | $D_1 \rightarrow D_2$ | $D_1 \rightarrow D_3$ | $D_2 \rightarrow D_3$ | $D_2 \rightarrow D_4$ | $D_3 \rightarrow D_4$ |
|-------|-----------|-----|-----|-----|-----|-----|
| Supervised | GAP | 24.9 | 59.1 | 5.8 | 830 | 888 |
| | SchNet | 13.2 | 15.4 | 6.2 | 93 | 107 |
| | SCN | 22.1 | 29.3 | 9.7 | 139 | 131 |
| Fine-tune | SchNet-FT | 17.6 | 27.3 | 3.7 | 121 | 116 |
| Ours | MEKRR-(SchNet) | 8.0 | 9.3 | 2.9 | **27** | 55 |
| | MEKRR-(SCN) | **7.0** | **6.3** | **2.0** | 40 | **42** |

## 5.5 Leveraging MEKRR beyond supervised learning

In the previous sections, we have shown how MEKRR performs very well on both supervised and transfer learning tasks. The effectiveness comes from the combination of a pre-trained feature map together with the $K_\alpha$ kernel. However, this idea is not restricted to supervised learning. We can indeed leverage the similarity measure provided by the kernel for tasks beyond potential energy regression. As a simple example, in Fig. 3 we plot the kernel matrix, when using SchNet as the feature map, of a part of the trajectory of $D_2$ containing an N-N bond breaking event in the cases $\alpha = 0$ and $\alpha = 1$. In both images we can see a clear structure that highlights at least two distinct states, but with the second heatmap having more signal. We can then use the kernel to perform spectral clustering with two classes, the result is visualized on the top margin of the heatmap along with the time evolution of a physical quantity that signals the N-N bond-breaking. This facilitates a

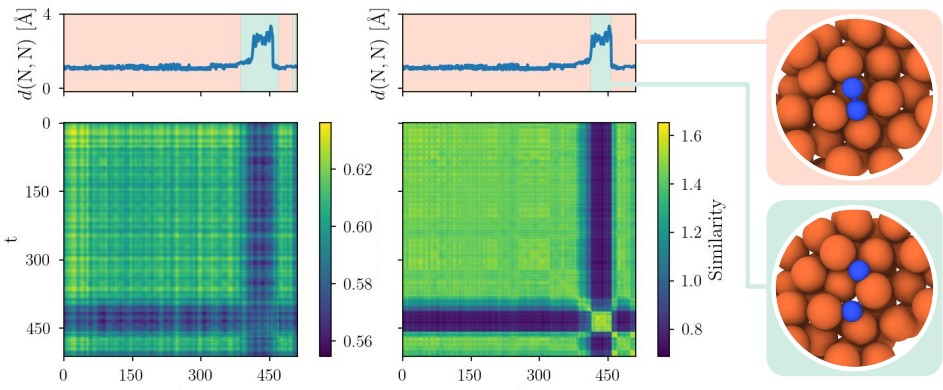

Figure 3: Heatmaps of the $K_\alpha$-SchNet kernel applied to a part of the trajectory of $D_2$ (where a reactive event occurs) and time series of the distance between nitrogen atoms over time $t$. The cases with $\alpha = 0$ and $\alpha = 1$ are reported on the left and right, respectively. Using spectral clustering with the two kernels as inputs we label each time-index with one of two classes, with the background color showing the class. Spectral clustering with the multi-weight kernel picks out the reactive event perfectly.

physical interpretation, as the two classes correspond to configurations containing the reactants (the $N_2$ molecule) and products (two N atoms) of the chemical reaction. Interestingly, the $\alpha = 1$ case correlates more closely with the handpicked physical quantity, shown in the top panel. The reason for this is the fact that learning the weights independently gives more weight to chemical species that are under-represented, which typically correspond to adsorbed atoms in surfaces. This allows us to give more weight to the most important actors in catalytic applications.

## 6 Conclusion and future work

In this work, we introduced an approach to model the potential energy surface of atomistic systems. Our method employs GNN representations trained on the large OC20 dataset along with kernel ridge regression, which we tested on two catalytic processes that are outside the distribution of the pre-training dataset. We devised a kernel function incorporating GNN features, blending kernel mean embedding with information related to the atom's chemical species. Our approach outperforms standalone GNN or kernel methods, demonstrating impressive transferability. This suggests promising avenues for transfer learning application in computational chemistry. However, we recognize certain limitations. Firstly, our method is based upon KRR which scales poorly to large scale datasets, although potential ways around this such as random features [70] or Nyström approximations [71] can overcome this limitation. Secondly, although we tested MEKRR on out-of-distribution datasets for representations pre-trained on OC20, we still focused our analysis on catalytic systems similar to those in the dataset. In this regard, it would be interesting to understand the extent to which MEKRR can predict well the chemical properties of generic systems. In addition, it would be important to incorporate forces into the loss function in order to use it in molecular dynamics applications. We believe that addressing these aspects will further improve the impact of this framework in computational chemistry.

**CRediT author statement**   **J. I. Falk**: Conceptualization, Methodology, Software, Investigation, Formal analysis, Writing - Original Draft; **L. Bonati**: Conceptualization, Methodology, Formal analysis, Visualization, Writing - Original Draft; **P. Novelli**: Conceptualization, Methodology, Writing - Original Draft; **M. Parrinello**: Conceptualization, Writing - Review & Editing, Supervision; **M. Pontil**: Conceptualization, Methodology, Writing - Review & Editing, Supervision.

**Acknowledgements**   We acknowledge the financial support from the PNRR MUR Project PE000013 CUP J53C22003010006 "Future Artificial Intelligence Research (FAIR)", funded by the European Union – NextGenerationEU, EU Project ELIAS under grant agreement No. 101120237, and the "Joint project TransHyDE_FP3: Reforming ammonia - transport of $H_2$ via derivatives", funded from the German Federal Ministry of Research (BMBF), funding code: 03HY203A-F.

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
