# Supplementary Material

This appendix is organized as follows: in Appendix A we derive the multi-weight KRR outlined in Section 4.3 in the main paper. In Appendix B we show times for training the algorithms and discuss this. In Appendix C we provide details on the algorithms and hyperparameters used in the paper, and conduct additional experiments to those in Section 5.

**Code** The code repository necessary to run the experiments are provided at the link `https://github.com/IsakFalk/atomistic_transfer_mekrr`.

## A Multi-weight KRR

In this section, we show that solving KRR with the chemically-informed mean embedding (5) is equivalent to optimize the objective function

$$\sum_{t=1}^{T} \left( E_t - \sum_{s=1}^{S} \langle w_s + w_0, \phi(H_s(x_t)) \rangle \right)^2 + \lambda \left( \frac{1}{\alpha} \sum_{s=1}^{S} \|w_s\|^2 + \frac{1}{1-\alpha} \|w_0\|^2 \right). \quad (6)$$

Our reasoning follows that in [57, 58] for the multi-task learning setting. Notably here, we work with embedding kernels and a different loss function, hence the final results is conceptually different.

We begin by defining the change of variable $u_s \leftarrow \frac{w_s}{\sqrt{\alpha}}, s \in [S]$, and $u_0 \leftarrow \frac{w_0}{\sqrt{1-\alpha}}$. Then let

$$\mathbf{u} = (u_0, u_1, \ldots, u_S), \quad \text{and} \quad \Psi(H(x_t)) = \left( \sqrt{1-\alpha}\phi(H(x_t)), \sqrt{\alpha}\phi(H_1(x_t)), \ldots, \sqrt{\alpha}\phi(H_S(x_t)) \right).$$

For any $\mathbf{u} = (u_0, u_1, \ldots, u_S)$ and $\mathbf{u}' = (u'_0, u'_1, \ldots, u'_S)$ we also define the inner product $\langle \mathbf{u}, \mathbf{u}' \rangle = \sum_{s=0}^{S} \langle u_s, u'_s \rangle$. With this notation the objective function (6) can be rewritten as

$$\sum_{t=1}^{T} \left( E_t - \langle \mathbf{u}, \Psi(H(x_t)) \rangle \right)^2 + \lambda \|\mathbf{u}\|^2$$

which we recognize as the usual KRR objective with RKHS given by the kernel in (5). Notice that this reasoning applies whenever $\alpha \in (0,1)$. The cases $\alpha = 0$ or $\alpha = 1$ can be treated following a similar reasoning. For instance, if $\alpha = 1$, since we aim to minimize the objective (6), we can drop the variable $w_0$ as at the optimum $w_0 = 0$.

## B Timings

Table 3: Timings (mean $\pm$ std) in seconds for training all algorithms using the same algorithmic settings as for Table 1 for $D_1, D_4$ and the Cu/formate datasets. MEKRR is much faster than the competitors despite being the most competitive in terms of performance (as seen in Table 1). The timings were aggregated over three independent runs. For each dataset, we specify the number of configurations $T$, atoms $n$ and species $S$ using a tuple $(T, n, S)$. The best performance given by **bold** number in gray cell.

| Algorithm | $D_1$ $(720, 47, 2)$ | $D_4$ $(600, 74, 2)$ | Cu/formate $(1000, 52, 4)$ |
|---|---|---|---|
| SchNet | $483 \pm 2$ | $512 \pm 5$ | $585 \pm 4$ |
| GAP-SOAP | $52 \pm 1$ | $69 \pm 1$ | $282 \pm 2$ |
| SchNet-FT | $442 \pm 1$ | $627 \pm 2$ | $708 \pm 1$ |
| MEKRR | $20 \pm 0.5$ | $38.5 \pm 0.5$ | $43 \pm 1$ |

As can be seen from Table 3, MEKRR is the fastest to train by a wide margin. While train time is dependent on both the size of the problem in terms of number of configurations $T$, number of atoms $n$ and number of chemical species $S$ and the specific hyperparameter used, this shows that for these settings, MEKRR performs well without sacrificing speed. A thing of note is that the train time of MEKRR is not drastically impacted by the number of chemical species $S$ compared to GAP-SOAP.

# C Experiments

## C.1 Hardware Specification

**OS** Ubuntu 20.04.6 LTS

**RAM** 128GB

**CPU** AMD Ryzen Threadripper PRO 5965WX 24-Cores

**GPU** NVIDIA GeForce RTX 3090

## C.2 Algorithm specification

We specify in more detail the algorithms and hyperparameter choices. For an exact specification, please inspect the supplied code base. All of the algorithms use periodic boundary conditions in all directions. For the same-system predictions for SchNet, SchNet-FT, SCN, and MEKRR we standardize the data, while for the transfer-system predictions, for all algorithms, we remove the mean using a precomputed dictionary $\bar{\epsilon} \in \mathbb{R}^S$ of the energies of each species $s$, so that the energy becomes $E \mapsto E - \sum_i^n \bar{\epsilon}_{z_i}$ and at test time we predict this residual and add back the correct sum.

**SchNet** We use the implementation of the OCP20 code base with the number of hidden channels being 256, number of filters being 64, number of interactions being 3 and number of Gaussians in the basis expansion being 200 and a cutoff for generating the graphs being 6.0Å. We perform optimization using the MSE objective with no regularization and use the AdamW optimizer with the amsgrad option and no weight decay with a learning rate of $10^{-4}$. We use a batch size of 16 and optimize for 800 epochs, saving the weights with the best validation error on the validation set using RMSE as the validation objective.

**SchNet-FT** We use the implementation of the OCP20 code base where we use the pretrained weights of `https://dl.fbaipublicfiles.com/opencatalystproject/models/2020_11/s2ef/schnet_all_large.pt` and config `https://github.com/Open-Catalyst-Project/ocp/blob/main/configs/s2ef/all/schnet/schnet.yml` (SchNet, trained on the split All in the OCP20 models table). This model has the hyperparameters of number of hidden channels being 1024, the number of filters being 256, the number of interactions being 5, number of Gaussians in the basis expansion being 200 and a cutoff for generating the graphs being 6.0Å. We freeze layers up to layer 3. We fit the remaining parameters by optimization using the MSE objective with no regularization and use the AdamW optimizer with the amsgrad option and no weight decay with a learning rate of $10^{-4}$. We use a batch size of 16 and optimize for 400 epochs, saving the weights with the best validation error on the validation set using RMSE as the validation objective.

**SCN** We use the implementation of the OCP20 code base with the number of interactions being 3, hidden channels being 64, sphere channels being 32, number of sphere samples being 128, `l_max` being 6 and number of bands being 2, the number of basis functions for SCN being 32 and a cutoff for generating the graphs being 6.0Å. We perform optimization using the MSE objective with no regularization and use the AdamW optimizer with the amsgrad option and no weight decay with a learning rate of $4 \cdot 10^{-4}$. We use a batch size of 16 and optimize for 800 epochs, saving the weights with the best validation error on the validation set using RMSE as the validation objective.

**GAP** As specified in the main body, we use the quippy python interface of the QUIP implementation of GAP. We set the following parameters in the command line interface `gap_fit` where we use the SOAP parameters `atom_sigma` $= 0.5$, `l_max` $= 6$, `n_max` $= 12$, `cutoff` $= 6.0$, `cutoff_transition_width` $= 1.0$, `delta` $= 0.2$, `covariance_type` $=$ `dot_product`, `n_sparse` $= 1000$, `zeta` $= 4$, `energy_scale` $= 1.0$, `atom_gaussian_width` $= 1.0$ and additional parameters of `default_sigma` $= [0.001, 0., 0., 0.]$, `e0_method` $=$ `average`, except for the transfer learning experiments where we instead remove the average using precomputed atom-specific energies as specified in the top paragraph of this section.

**MEKRR** For MEKRR-(SchNet) we use the same pretrained weights and configuration as those of SchNet-FT above, and we extract the representation as the output of the second layer.

For MEKRR-(SCN), we use the implementation of the OCP20 code base where we use the pretrained weights of `https://dl.fbaipublicfiles.com/opencatalystproject/models/2023_03/s2ef/scn_all_md_s2ef.pt` and config `https://github.com/Open-Catalyst-Project/ocp/blob/main/configs/s2ef/all/scn/scn-all-md.yml` (SCN, trained on the split All+MD in the OCP20 models table). This model has the hyperparameters of number of hidden channels being 1024, number of sphere channels being 128, the number of interactions being 16, number of Gaussians in the basis expansion being 200 and a cutoff for generating the graphs being 6.0Å. For additional hyperparameters, see the configuration file at `https://github.com/Open-Catalyst-Project/ocp/blob/main/configs/s2ef/all/scn/scn-all-md.yml`. The GNN feature map is obtained as the output of the 8th layer of SCN. Furthermore, in order to get invariant feature vectors from the $C$ channels of spherical harmonics functions of the $L$'th layer, $(s_{L,c})_{c=1}^C$ we perform a reduction $s_{L,c} \mapsto h_{L,c} = \int s_{L,c}(r)\mathrm{d}r$ and stack them into a feature vector $h_L = (h_{L,c})_{c=1}^C$.

For the datasets in Table 1 we use $\lambda = 10^{-7}$ and $\alpha = 1.0, 10^{-2}$ for the Fe/N$_2$ and Cu/Formate datasets, respectively with $C = 1/n$. For the datasets in Table 2 we use $\lambda = 10^{-4}$ and $\alpha = 0.0$.

In order to transfer between different system sizes (as in Table 2) we note that, if the model has been trained with data $(H(x_t))_{t=1}^T$ from a system of given size, and one wants to estimate the energy $E(H)$ of a configuration $H$ belonging to a system of different size $n_H$ whose node embeddings are $(h_i)_{i=1}^{n_H}$, one has

$$E(H) \approx f(H) = \sum_{i=1}^{n_H} \sum_{t=1}^{T} c_t \langle \phi(h_i), \phi(H(x_t)) \rangle$$

where $(c_t)_{t=1}^T$ are the fitted coefficients for MEKRR. For this reason, in this setting we use $C$ equal to 1.

## C.3   Additional experiments

All of the below experiments are done with the pre-trained SchNet feature map.

**Necessity of non-linear kernel in MEKRR**    To show that averaging the node-features from the pre-trained SchNet GNN fails to learn we apply MEKRR with a linear kernel to the output of the second layer to the pretrained SchNet. The results are shown in Table 4. We can clearly see that Linear-KRR (MEKRR with a linear kernel applied to SchNet) fails to perform well highlighting the need for a non-linear kernel for MEKRR to work.

**Representation layer**    In all experiments of MEKRR-(SchNet) we construct the feature vector as the output of the second layer of the pre-trained model. This is motivated by the tradeoff between memory, computation and performance. In this section we report the results when varying the number of layers on the same-energy prediction on dataset $D_2$ (Table 5) for the two limiting case ($\alpha = 0$ and $\alpha = 1$). In both cases, there is only a slight increase in performance. Furthermore, we report the visualizations of the kernel matrix in Fig. 4. For both tasks, we observe an improvement when using the chemical species-informed MEKRR variant.

**Evaluation tables**    In Table 6 and Table 7 we report additional metrics for the same-dataset and across-datasets energy predictions. We also distinguish the general multispecies formulation from that which uses shared weights ($\alpha = 0$). Note that for the Fe/N$_2$ datasets, the value of $\alpha$ is not cross-validated on every dataset but only on $D_2$ (see Section 5.3).

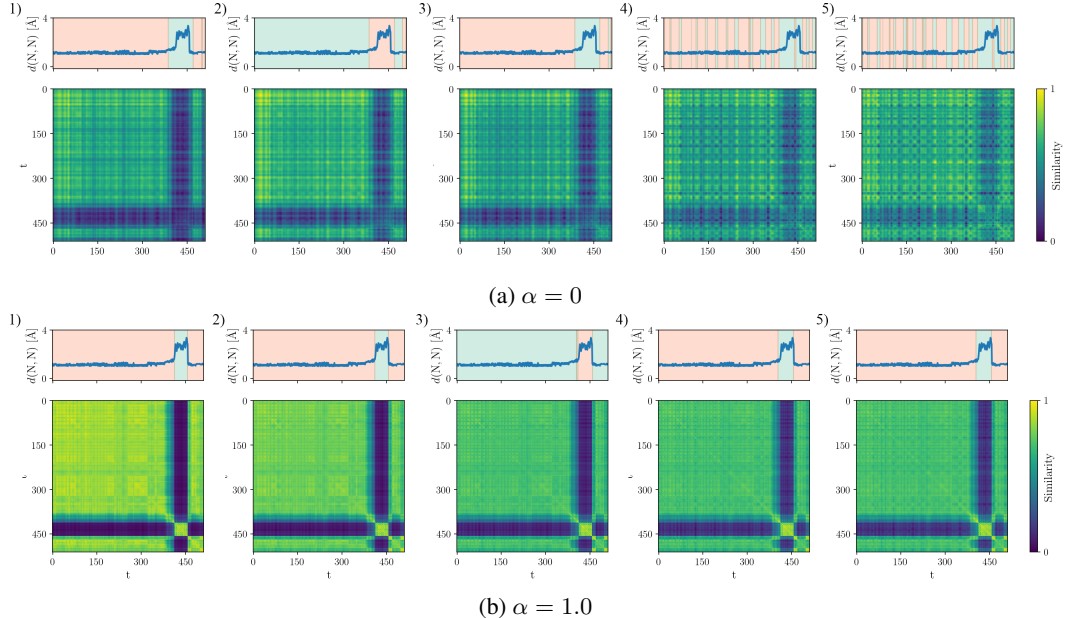

(a) $\alpha = 0$

(b) $\alpha = 1.0$

Figure 4: Heatmaps of the $K_\alpha$ kernel applied when $k$ is Gaussian (lengthscale fit with median heuristic) and the node-features are given by the $L$'th layer $L = 1, \ldots 5$ of the pretrained SchNet GNN going from the leftmost to rightmost column, to a part of the trajectory of $D_2$ (where a reactive event occurs) and time series of the distance between nitrogen atoms over time $t$. The cases with $\alpha = 0$ and $\alpha = 1$ are reported above and below, respectively. Using spectral clustering with the two kernels as inputs we label each time-index with one of two classes, with the background color showing the class. The color has been normalized to be between 0 and 1 which does not affect the clustering or visualization. We can see that output representation from later layers yield more patterned kernel matrices with more erratic clustering. Using the multi-weight kernel $K_\alpha$ where $\alpha = 1$ gives better results across the board.

Table 4: Comparison between linear and Gaussian kernel, same-dataset energy prediction for the Fe/N$_2$ datasets $D_i$. The errors are in units of meV/atom. Best performance given by **bold** number in gray cell.

| Algorithm | D1 | | D2 | | D3 | | D4 | |
|---|---|---|---|---|---|---|---|---|
| | RMSE | MAE | RMSE | MAE | RMSE | MAE | RMSE | MAE |
| Linear-KRR-($\alpha = 0$) | 72 | 69 | 198 | 166 | 66 | 53 | 146 | 118 |
| Linear-KRR-($\alpha = 1$) | 21 | 18 | 155 | 125 | 161 | 128 | 55 | 40 |
| MEKRR-($\alpha = 0$) | 0.3 | 0.3 | 1.5 | 1.2 | **2.2** | **1.7** | **2.2** | **1.8** |
| MEKRR-($\alpha = 1$) | **0.1** | **0.1** | **1.3** | **0.9** | 2.4 | **1.7** | 3.3 | 2.3 |

Table 5: Comparison of MEKRR using different representation layers for the feature map, tested on the same-dataset energy prediction task for the Fe/N$_2$ dataset $D_2$. The errors are in units of meV/atom. Best performance given by **bold** number in gray cell.

| Layer | $\alpha = 0$ | | $\alpha = 1$ | |
|---|---|---|---|---|
| | RMSE | MAE | RMSE | MAE |
| 1 | 1.9 | 1.4 | 1.7 | 1.2 |
| 2 | 1.5 | 1.1 | 1.3 | 0.9 |
| 3 | 1.4 | 1.0 | 1.0 | 0.7 |
| 4 | 1.4 | 1.0 | 1.0 | 0.7 |
| 5 | 1.4 | 1.0 | 0.9 | 0.6 |

Table 6: Same-dataset energy prediction. The errors are in units of meV/atom. Best performance given by **bold** number in gray cell. With respect to Table 1, we also report the Mean Absolute Error (MAE) metric and we also add the case with $\alpha = 0$.

| Algorithm | Fe/N$_2$ | | | | | | | | Cu/formate | |
|---|---|---|---|---|---|---|---|---|---|---|
| | $D_1$ | | $D_2$ | | $D_3$ | | $D_4$ | | | |
| | RMSE | MAE | RMSE | MAE | RMSE | MAE | RMSE | MAE | RMSE | MAE |
| GAP | 0.4 | 0.4 | 2.1 | 1.5 | 3.9 | 2.9 | 4.9 | 3.0 | 2.8 | 1.4 |
| SchNet | 0.5 | 0.4 | 4.1 | 3.2 | 5.1 | 3.8 | 6.2 | 4.7 | 6.0 | 4.7 |
| SCN | 0.3 | 0.2 | 5.1 | 3.8 | 7.5 | 5.9 | 7.3 | 5.8 | 2.5 | 2.0 |
| SchNet-FT | **0.1** | **0.1** | 2.0 | 1.5 | 2.5 | 3.2 | 3.2 | 2.6 | 1.9 | 1.5 |
| MEKRR-(SchNet)-0 | 0.3 | 0.3 | 1.5 | 1.2 | 2.2 | 1.7 | **2.2** | **1.8** | 1.7 | 0.9 |
| MEKRR-(SCN)-0 | 0.2 | 0.2 | 1.8 | 1.4 | 3.6 | 2.9 | 6.8 | 5.2 | 1.8 | **0.6** |
| MEKRR-(SchNet) | **0.1** | **0.1** | 1.3 | 0.9 | 2.4 | 1.7 | 3.3 | 2.3 | **1.2** | **0.6** |
| MEKRR-(SCN) | 0.2 | **0.1** | **0.9** | **0.7** | **1.9** | **1.5** | 2.7 | 2.1 | 1.7 | **0.6** |

Table 7: Transfer evaluation of algorithms on source to target: $D_{\text{source}} \rightarrow D_{\text{target}}$. The errors are in units of meV/atom. Best performance given by **bold** number in gray cell. With respect to Table 2, we also report the Mean Absolute Error (MAE) metric.

| Algorithm | $D_1 \rightarrow D_2$ | | $D_1 \rightarrow D_3$ | | $D_2 \rightarrow D_3$ | | $D_2 \rightarrow D_4$ | | $D_3 \rightarrow D_4$ | |
|---|---|---|---|---|---|---|---|---|---|---|
| | RMSE | MAE | RMSE | MAE | RMSE | MAE | RMSE | MAE | RMSE | MAE |
| GAP | 24.9 | 14.6 | 59.1 | 34.1 | 5.8 | 4.2 | 830 | 829 | 888 | 888 |
| SchNet | 13.2 | 10.1 | 15.4 | 12.3 | 6.2 | 4.9 | 93 | 90 | 107 | 105 |
| SCN | 22.1 | 18.3 | 29.3 | 23.1 | 9.7 | 7.7 | 139 | 136 | 131 | 129 |
| SchNet-FT | 17.6 | 13.6 | 27.3 | 19.4 | 3.7 | 2.8 | 121 | 119 | 116 | 114 |
| MEKRR-(SchNet) | 8.0 | 5.6 | 9.3 | 6.9 | 2.9 | 2.2 | **27** | **20** | 55 | 51 |
| MEKRR-(SCN) | **7.0** | **4.8** | **6.3** | **5.1** | **2.0** | **1.6** | 40 | 32 | **42** | **34** |