# OpenReview forum: "Transfer learning for atomistic simulations using GNNs and kernel mean embeddings"
_NeurIPS.cc/2023/Conference — NeurIPS 2023 poster_

### Official Review · Reviewer_f9vG · 2023-07-01

**Soundness:** 3 good
**Presentation:** 3 good
**Contribution:** 3 good
**Rating:** 7
**Confidence:** 3

**Summary:**

In this work, the authors investigate the use of kernel ridge regression with feature maps of atomic environments from pre-trained graph neural network models for predicting energies of atomistic systems. They show that their method, mean embedding kernel ridge regression (MEKRR), outperforms or is competitive with pre-trained and fine-tuned SchNet models trained on the OC20 dataset. Their kernel-based method is interpretable and gives insight into the chemical evolution of systems in trajectories.

**Strengths:**

The paper is well written and clear, giving a nice overview of both GNN- and kernel-based approaches to modeling potential energy surfaces.

The method is explained well and well-motivated by the outstanding problems related to transferability in the field of machine learned interatomic potentials.


**Weaknesses:**

The authors only evaluate their method against one GNN, SchNet; there is a whole zoo of modern GNNs purpose-built for learning potential energy surfaces that incorporate additional physical priors and are fast and easy to train.

One key issue with neural interatomic potentials is transferability across geometric environments; it is not clear that MEKRR addresses this challenge in any way, as it is ultimately reliant on pre-computed feature maps.


**Questions:**

Do the authors think that scalability will be an issue with their method in practical workflows, i.e. for the million - billion scale atomistic simulations for solid state and biomolecular systems that are now being done with neural interatomic potentials?

Why is SchNet used as a baseline rather than Allegro, MACE, Spherical Channel Network, or some of the many other GNNs that are commonly used to model the OC20 dataset? What advantages does MEKRR provide over modern GNN approaches that can be quickly and accurately fine tuned?


**Limitations:**

Limitations are addressed in part, but further discussion is needed related to more modern ML interatomic potentials.

---

> ### Author Rebuttal · Authors · 2023-08-09
>
> > The authors only evaluate their method against one GNN, SchNet; there is a whole zoo of modern GNNs purpose-built for learning potential energy surfaces that incorporate additional physical priors and are fast and easy to train.
>
> As discussed above we do not simply test our method against a GNN, but we synergistically combine the representation power of GNN with the simplicity of kernel ridge regression. To further support the generality of our approach, we added new experiments (see Table 1 and Table 2) in which we both extract the feature map from SCN and use this method as a baseline.
>
> > One key issue with neural interatomic potentials is transferability across geometric environments; it is not clear that MEKRR addresses this challenge in any way, as it is ultimately reliant on pre-computed feature maps.
>
> This is not correct. Indeed, the experiment in Table 2 measures exactly the transferability across geometric environments. Indeed, each of the $D_1$, … , $D_4$ datasets cover increasingly larger regions of the phase space (see detailed description in section 5.2). MEKRR performing better on this task is a clear sign of improved transferability than using GNN methods alone. This is another key contribution of our work: to show that the representation learned from the OC20 dataset (consisting of a few correlated samples of many chemical systems) can provide an effective representation also for out-of-sample configurations of chemical interest. The datasets tested, which are taken from real-life applications to molecular dynamics, indeed contain reactive configurations in which bonds are broken or formed which are severely out-of-sample for the pre-trained feature map.
>
> > Do the authors think that scalability will be an issue with their method in practical workflows, i.e. for the million - billion scale atomistic simulations for solid state and biomolecular systems that are now being done with neural interatomic potentials?
>
> If scalability refers to predicting energy for systems with large numbers of atoms, we see no obstacles in principle. In fact, recent work [1] has shown how, with proper adjustments, simulations of 0.5 trillion atoms can be performed using a Gaussian process, which shares the same limitations as our method.
>
> > Why is SchNet used as a baseline rather than Allegro, MACE, Spherical Channel Network, or some of the many other GNNs that are commonly used to model the OC20 dataset? What advantages does MEKRR provide over modern GNN approaches that can be quickly and accurately fine tuned?
>
> The goal of the paper is not to perform well on OC20, but how to effectively transfer GNNs pretrained on OC20 to systems qualitatively different from OC20, see the general answer and the final comment to Rev. D8fw concerning evaluation criteria. As suggested by the reviewer, we have added SCN to the experiments (Table 1 and Table 2) as a baseline, GNN fine-tuning and a feature map for MEKRR. We chose SCN since it was implemented in the OCP codebase and thus easy to integrate into our codebase and had pretrained weights available.
>
> Regarding fine-tuning of a pretrained GNN representation with MEKRR vs standard fine-tuning, we invite the reviewer to look at Table 3, in Appendix, section B “Timings”, for some wallclock comparison of computational complexity of training a GNN from scratch vs fine-tuning vs MEKRR where MEKRR is order of magnitudes faster. Indeed, for the new SCN pre-trained model we were not able to fine-tune it to the new dataset with an acceptable quality, see the general answer above.
>
> Furthermore, in realistic applications one often has to resort to active learning strategies, and retrain the potential whenever new configurations are labeled. In this scenario, kernel ridge regression methods are very appealing.
>
> > Limitations are addressed in part, but further discussion is needed related to more modern ML interatomic potentials.
>
> Following your and other reviewers' suggestion we will include in the revised manuscript an expanded discussion on limitations.
>
> [1] Johansson et. al., arXiv:2204.12573 (2022)

---

> > ### Comment · Reviewer_f9vG · 2023-08-11
> >
> > I have read the rebuttal and thank the authors for their detailed response. In light of the new experiments with SCN and their responses to my comments, I will happily revise my score and recommend acceptance.

---

> > > ### Author Response · Authors · 2023-08-14
> > > **Thank you**
> > >
> > > We thank the reviewer for raising their score and recommending acceptance, in addition to the kind words and constructive suggestions which will help us improve our work. We would like to remind the reviewer to increase the score when possible.

---

### Official Review · Reviewer_D8fw · 2023-07-06

**Soundness:** 2 fair
**Presentation:** 2 fair
**Contribution:** 2 fair
**Rating:** 5
**Confidence:** 3

**Summary:**

The paper proposes a transfer learning algorithm that leverages the ability of graph neural networks (GNNs) together with kernel mean embeddings. A flexible kernel function that incorporates chemical species information is proposed to improve performance and interoperability. Empirical results in a series of realistic datasets of increasing complexity are provided to illustrate the generalization and transferability performance of the proposed method, MEKRR.

**Strengths:**

* MEKRR achieves excellent generalization and transferability performance on the introduced datasets with increasing complexity.
* The paper is well-organized and clearly written.

**Weaknesses:**

* The proposed mean embedding kernel ridge regression (MEKRR) is claimed to have better generalization and transferability, however, no substantial theoretical evidence or intuition is provided to motivate this claim. Consequently, it is unclear why MEKRR could achieve better performance than other methods.
* MEKRR cannot directly calculate the molecular force as the gradient flow of pre-trained feature maps is stopped, which limits its application in real-world scenarios.
* In inference, it appears that kernel-based methods such as MEKRR usually need to evaluate the kernel function of a target molecule with all molecules in the training set. This linear scaling computational cost can be inefficient for large-scale datasets, in contrast to most GNN-based methods, which have a fixed inference complexity.
* Concerns exist regarding experimental evaluation:
    * The molecule scales in the proposed datasets are relatively small, making it difficult to justify the generalization capacity of MEKRR on a broader range of large molecules. Additionally, the limited data size negatively affects the performance of GNNs, meaning that comparisons on small-scale datasets may not adequately reveal the difference between MEKRR and SchNet.
    * Comparison with only the SchNet model is insufficient, as SchNet is not the state-of-the-art (SOTA) model in the S2EF task of the OCP20 dataset [https://github.com/Open-Catalyst-Project/ocp/blob/main/MODELS.md]. It is also unclear whether SchNet is representative enough to benchmark the generalization and transferability of those SOTA models. Thus, it is recommended that the authors include more pretrained GNN models to further verify the effectiveness of MEKRR.
    * The criterion for evaluating the generalization and transferability is unclear, making it difficult to justify the significance of the empirical results. As shown in Table 1-2, although MEKERR exhibits better performance than the baseline methods, there exist large gaps in RMSE/MSE metrics between in-distribution and out-of-distribution settings. For instance, RMSE in $D_1\rightarrow D_2$ is 80 times larger than RMSE in $D_1$. Is the scale of RMSE in $D_1\rightarrow D_2$ acceptable in real chemistry research?

**Questions:**

* In the main text, Equation 7 mentioned in line 205 is not found. Could the authors clarify this further?
* Establishing MEKRR only on SchNet does not adequately support the universality of the proposed algorithm. It is recommended that the authors include more pretrained GNN models in the experimental evaluation.
* Could the authors provide more clarification about the criterion for evaluating the generalization and transferability of a machine learning-based PES model? This would make the contribution of MEKRR clearer.

**Limitations:**

The authors well clarify the limitations and potential negative societal impact of their work in the paper.

---

> ### Author Rebuttal · Authors · 2023-08-09
>
> > … no substantial theoretical evidence or intuition is provided to motivate this claim.
>
>
> While we do not provide formal evidence we believe the experimental results speak for themselves. Theoretical justification could be made using results from transfer learning and kernel ridge regression. A good starting point could be the work by [1].
>
> > MEKRR cannot directly calculate the molecular force as the gradient flow of pre-trained feature maps is stopped, which limits its application in real-world scenarios.
>
> This is not correct. Although in this paper for simplicity we only focused on energy prediction, the forces can be simply calculated via the chain rule, multiplying the derivatives of the kernel with respect to the feature vector by the derivatives of the feature vector with respect to atomic positions; see [2] for how to do this effectively in practice using automatic differentiation.
>
> > This linear scaling computational cost can be inefficient for large-scale datasets, in contrast to most GNN-based methods, which have a fixed inference complexity.
>
> First, we would like to stress that our focus is to perform transfer learning to a target atomistic system, hence the training set will typically be rather small. Indeed, datasets used for modeling realistic potentials for molecular dynamics applications are of the order of $10^3-10^4$ samples.
>
> Second, there are several techniques that can be used to remove the linear scaling of kernel methods (such as random fourier features, Nystrom sampling etc.) which could be studied in future work. We will add this comment in the discussion of the limitations.
>
> > The molecule scales in the proposed datasets are relatively small, making it difficult to justify the generalization capacity of MEKRR on a broader range of large molecules. Additionally, the limited data size negatively affects the performance of GNNs, meaning that comparisons on small-scale datasets may not adequately reveal the difference between MEKRR and SchNet.
>
> We disagree, and indeed, this is exactly the point of our work: to show that the GNN-based representations trained in OC20 transfer well to small datasets when used with kernel mean embeddings and kernel ridge regression. As we discuss in the general answer, we consider a real-life scenario in which we want to specialize our pre-trained model to a new system, using a transfer learning approach. This is motivated by the fact that labeling of new configurations is very expensive and so in the practice one has to work with the constraint of limited data availability.
>
> For this reason, we should test the generalization ability of the model not to larger molecules but rather to different regions of the configurational space, which is exactly what we do in the transfer-learning experiments reported in Table 2.
>
> Furthermore, our work is in synergy and not in competition with the GNN approaches. Indeed, it is thanks to the representations learned by GNN architectures that a conceptually simple method like kernel ridge regression works so well!
>
> > In the main text, Equation 7 mentioned in line 205 is not found. Could the authors clarify this further?
>
> This should be Equation (4) (see the supplementary material for the corrected cross-reference). We will update this in the final version of the paper.
>
> > Establishing MEKRR only on SchNet does not adequately support the universality of the proposed algorithm.  It is recommended that the authors include more pretrained GNN models in the experimental evaluation.
>
> As described in the general answer, we repeated the experiments using SCN, an equivariant model which was devised specifically for the OC20 dataset and indeed it is among SOTA models. As discussed in the general answer (see Tables 1 and 2) all the conclusions obtained when using SchNet hold also for this more elaborate GNN model, supporting the generality of the proposed approach.
>
> > … Could the authors provide more clarification about the criterion for evaluating the generalization and transferability of a machine learning-based PES model? This would make the contribution of MEKRR clearer.
>
> As discussed above, a crucial property is transferability across geometric environments. To this extent, we consider the transfer learning across the $D_i$ datasets. They have been taken from a single dataset used to study a realistic catalytic application, and divided according to the sampling technique used, covering an increasingly larger portion of the configuration space. As detailed in section 5.2, they represent by construction complementary datasets of increasing complexity, for which it is natural to obtain gaps in validation metrics between in-distribution and out-of-distribution settings.
>
> The fact that MEKRR obtains consistently better results on this task (sometimes by one order of magnitude) is a clear indication of its better transferability to wider portions of the configuration space.
>
> > … For instance, RMSE in D1→D2  is 80 times larger than RMSE in D1. Is the scale of RMSE in D1→D2 acceptable in real chemistry research?
>
> $D_1$ represents a limiting case, where the configurations are collected via DFT (oracle)-based molecular dynamics, and as such they are composed of very correlated and near-equilibrium. configurations, meaning that the effective sample size is very small. Indeed, collecting configurations only from ab initio molecular dynamics is not sufficient [4] and needs to be complemented with other sampling schemes. We will expand the discussion of the datasets clarifying this.
>
> [1] Tripuraneni, Jordan, and Chi Jin, Advances in neural information processing systems 33 (2020): 7852-7862.
>
> [2] Schmitz, Klaus-Robert Müller, and Chmiela, The Journal of Physical Chemistry Letters 13.43 (2022): 10183-10189.
>
> [3] Chatalic, Antoine, et al, International Conference on Machine Learning. PMLR, 2022.
>
> [4] Unke, Oliver T., et al., Chemical Reviews 121.16 (2021): 10142-10186.

---

> > ### Comment · Reviewer_D8fw · 2023-08-12
> > **Response**
> >
> > * Could the authors provide further clarification on the intuition or theoretical evidence as to why MEKRR achieves better performance than fine-tuning GNN? I believe this is necessary for inspiring future works following this paper and understanding the transferability challenges in ML-based PES models.
> > * Are the authors able to present any results regarding force prediction on some datasets, if the cost is reasonable?
> > * Regarding the fourth question mentioned above, could the authors provide an approximate or estimated time comparison between labeling new configurations using DFT and performing inference with a GNN/MEKRR model on the considered dataset? This information would be helpful in supporting the necessity of such a transfer learning setting.

---

> > > ### Author Response · Authors · 2023-08-14
> > > **Point-by-point response**
> > >
> > >
> > > > Could the authors provide further clarification on the intuition or theoretical evidence as to why MEKRR achieves better performance than fine-tuning GNN? I believe this is necessary for inspiring future works following this paper and understanding the transferability challenges in ML-based PES models.
> > >
> > > It is well-known that kernel methods are more efficient than NNs in low-data regimes: kernel methods are universal approximators [A] in addition to being non-parametric which allows the complexity of the model to grow with the size of the train set [B] while retaining generalization guarantees [C]. Specifically for MEKRR, mean embeddings allows us to define a metric on distributions which separates any distributions which are different. In total, MEKRR is a principled way to learn over point clouds. From a practical perspective, kernel ridge regression is much easier to train than neural networks for small to medium-sized datasets (since they solve a strongly convex problem and have a unique solution) with less hyperparameters to tune.
> > >
> > > On the other hand, kernel methods notoriously fall short on two key aspect:
> > > 1. Unfavorable scaling of the computational complexity for large datasets,
> > > 2. Off-the-shelf kernels beyond vectorial (unstructured) data working less well
> > >
> > > While we believe 1. is really not an issue in the transfer learning pipelines we foresee, where the datasets are small-to-medium sized (i.e. smaller than few ten of thousands points) and can be mitigated using e.g. Nystrom or random fourier features (see previous response on linear scaling),  in the spirit of previous meta-learning works, we proposed the use of a pre-trained representation to overcome the difficulty posed by 2. Thus our approach combines the representation power of large-scale trained GNNs with the inductive bias of KRR.
> > >
> > > > Are the authors able to present any results regarding force prediction on some datasets, if the cost is reasonable?
> > >
> > > We highlight that our method currently can produce force predictions at a reasonable additional cost, analogous to GNNs, and note that our repository __already implements force prediction__ for MEKRR fit to energies. However, at present it is difficult to incorporate forces with MEKRR during __training__ due to missing auto-diff primitives in the torch ecosystem and for this reason we have not presented results on force prediction. In detail, to fit MEKRR to forces, we have terms of the form $\frac{\partial K(x, x’)}{\partial R_{tij}} c_{tij}$ [42, submission] which we need to evaluate for all $t, i, j$ tuples going from $1$ to $T, n, 3$ respectively (number of frames, atoms, and dimensions of space), which is currently not possible in a vectorized manner.  Note that an implementation of an algorithm similar to us in [37, submission] has been implemented in Jax which we will consider in the future to allow fitting forces. We choose to use torch due to the OCP repository being implemented in this framework which allowed for easier loading of pretrained models. Another possibility for side-stepping the technical difficulties would be to modify the KRR approach by for example introducing random fourier features to approximate the Gaussian kernel and get an approximate finite-dimensional feature map.
> > >
> > > We will expand the conclusion section with these considerations.
> > >
> > > > Regarding the fourth question mentioned above, could the authors provide an approximate or estimated time comparison between labeling new configurations using DFT and performing inference with a GNN/MEKRR model on the considered dataset? This information would be helpful in supporting the necessity of such a transfer learning setting.
> > >
> > > The time needed to label a new configuration with DFT for the small system (47 atoms) is approximately 900 seconds. On the other hand, the time needed to perform inference with the GNN/MEKRR models, is approximately 0.1 seconds (GNN-FT and MEKRR being approximately the same) showing a difference of 4 orders of magnitude. The disparity of time scales explains the need to work with small datasets and to use a transfer learning setting. We will add this information in the timings section of the appendix.
> > >
> > > [A] C. A. Micchelli, Y. Xu, H. Zhang. Universal Kernels. J. of Machine Learning Research, 2006.
> > >
> > > [B] A. Caponnetto and E. De Vito. Optimal rates for the regularized least-squares algorithm. Foundations of Computational Mathematics 7:331-368, 2007.
> > >
> > > [C] C. J. Simon-Gabriel and B. Schölkopf. Kernel distribution embeddings: Universal kernels, characteristic kernels and kernel metrics on distributions. J. of Machine Learning Research 19:1708-1736, 2018.

---

> > > > ### Comment · Reviewer_D8fw · 2023-08-15
> > > > **Response**
> > > >
> > > > According to the response, I will raise my score.

---

### Official Review · Reviewer_qEyK · 2023-07-08

**Soundness:** 4 excellent
**Presentation:** 3 good
**Contribution:** 3 good
**Rating:** 7
**Confidence:** 5

**Summary:**

The authors propose to use embedding of GNNs pretrained on large and diverse molecular datasets (in their case OC20) to construct kernel embeddings for efficient transfer learning.

**Strengths:**

This paper was a delight to read. It solves a real problem with a clever idea, implemented well, and I learned something. The presentation is excellent and I have to say, it is incredibly refreshing to read a NeurIPS paper on AI4Science by authors that clearly understand the physics and chemistry behind what they are doing.

**Weaknesses:**

The authors mention meta-learning a few times in the intro section, but meta-learning is a big field and can mean many things. This is hard to understand for the reader without looking up the references.

Given the success of equivariant models over invariant ones like SchNet it would super interesting to see how to extend this approach to a setting where the kernel embedding is constructed from embeddings made of a direct sum of irreducible representations. The baselines chosen here are rather weak and were maybe state-of-the-art 5 years ago (I understand that pretrained equivariant potentials on OC20 may not have been available when the authors started this project and would've been computationally difficult to obtain).

**Questions:**

N/A

---

> ### Author Rebuttal · Authors · 2023-08-09
>
> Thanks for your positive comments.
>
> > The authors mention meta-learning a few times in the intro section, but meta-learning is a big field and can mean many things. This is hard to understand for the reader without looking up the references.
>
> We will clarify this point further in our revision. Our method is similar in spirit to meta-learning a data representation which is fine tuned on novel tasks. A core insight behind meta-learning  comes from few-shot image classification (e.g. refs. [29–33] in the submission), where we want to learn an algorithm that adapts quickly to new tasks given a collection of image classification tasks, and  an integral part of this approach is to add a preprocessing step where each image is mapped through a meta-learned or pre-trained representation.
>
> > …it would super interesting to see how to extend this approach to a setting where the kernel embedding is constructed from embeddings made of a direct sum of irreducible representations [...] the baselines chosen here are rather weak and were maybe state-of-the-art 5 years ago
>
> We have added an experiment using an equivariant model (SCN), see the general answer above in addition to Tables 1 and 2 in the pdf. A more formal investigation of how to use equivariant representations in the MEKRR algorithm (e.g. when dealing with the direct sum of irreducible representations as atom embeddings) is an important direction but outside of the scope of the current paper.

---

> > ### Comment · Reviewer_qEyK · 2023-08-11
> >
> > I have read the rebuttal and continue to recommend acceptance.

---

> > > ### Author Response · Authors · 2023-08-14
> > > **Thank you**
> > >
> > > We thank the reviewer once again for the positive comments and constructive suggestions which will allow us to improve our work.

---

### Official Review · Reviewer_7Ada · 2023-07-28

**Soundness:** 2 fair
**Presentation:** 2 fair
**Contribution:** 3 good
**Rating:** 6
**Confidence:** 3

**Summary:**

This paper presents a transfer learning algorithm for interatomic potentials, leveraging GNNs and kernel mean embeddings. It overcomes data and computational challenges, achieving generalization and transferability on realistic datasets.

**Strengths:**

The benchmark provided in this paper is relatively comprehensive. However, to ensure reproducibility, it would be ideal for the authors to make the complete code available.

**Weaknesses:**

The authors utilize pre-trained GNN embeddings but only discuss a limited number of SOTA GNN methods in atomistic simulations. It is crucial for the paper to clearly highlight the novelty and enhancements of their proposed method compared to existing GNN works for atomistic simulations [1]. Additionally, conducting an ablation study in the experimental section would be essential to demonstrate the innovative aspects of their approach. Simply applying a regular GNN method does not suffice as a significant contribution.

The methodology section seems odd as it contains only one subsection (4.1) despite the authors' claim of multiple contributions. It is crucial for the authors to explicitly clarify which aspects of their work are entirely new methods and which are applications of existing methods. This distinction will help readers better understand and appreciate the original contributions made in the paper.


[1] ​​Sriram, Anuroop, et al. "Towards training billion parameter graph neural networks for atomic simulations." arXiv preprint arXiv:2203.09697 (2022).


**Questions:**

Could the inclusion of a flowchart improve readers' ability to follow the pipeline of this paper? I found some difficulty in comprehending the research process while reading through the manuscript.

In the motivation, the authors mentioned the importance of rotational and translational invariance. Therefore, it would be relevant to discuss methods like SE3-equivariant approaches that handle these invariances.


**Limitations:**

The paper lacks in-depth discussions on limitations, such as issues regarding model robustness and ethical considerations, particularly regarding data misuse. It would be beneficial for the authors to address these aspects, as they are crucial in understanding the broader implications and potential challenges associated with the proposed models. Including a thoughtful exploration of these topics would enhance the completeness and relevance of the paper.

---

> ### Author Rebuttal · Authors · 2023-08-09
>
> >…it would be ideal for the authors to make the complete code available.
>
> The link to the code can be found in the supplementary material (https://anonymous.4open.science/r/mekrr-76F1/).
>
> > Novelty of the proposed method and comparison to to existing GNN works…
>
> We will expand the contribution section with the comments made in the global reply above, highlighting the key points of novelty. Nevertheless, we wish to point out that [1] concerns parallelizing GNN training which is not the aim of this work (we run all of our experiments using 1 GPU).
>
> > Additionally, conducting an ablation study in the experimental section would be essential…
>
> We have provided an ablation study in the Appendix, C.3. We note that since the GNN feature map is fixed there is nothing to ablate with respect to it. Instead we focus on the kernel embedding part of the pipeline, showing that the Gaussian kernel is integral to good performance by comparing it with the linear kernel on the same GNN as the feature map. Finally we include a study on how the representation layer of the GNN affects performance.
>
> > The methodology section seems odd as it contains only one subsection (4.1) [...]despite the authors' claim of multiple contributions. It is crucial for the authors to explicitly clarify which aspects of their work are entirely new methods and which are applications of existing methods. This distinction will help readers better understand and appreciate the original contributions made in the paper.
>
> We will rewrite the contributions section building on the paragraph in the global response. In particular, we will stress that we are **not** proposing a new GNN architecture for atomistic systems, but rather our goal is transferring pretrained GNNs to new systems effectively using a novel combination with kernel mean embeddings.
>
> Additionally, we will reorganize the methods section to reflect the inclusion of equivariant models in the experiments. In section 4.1 we will first describe the extraction of the feature vectors from GNN node features of invariant models like SchNet as well as from equivariant ones like SCN. Then in section 4.2 we will discuss the MEKRR method to fine tune the representation on a new dataset, and finally in section 4.3 we will introduce the more general version with a chemical-species dependent term.
>
> > Could the inclusion of a flowchart improve readers' ability to follow the pipeline of this paper?
>
> Thanks for the suggestions, we will prepare such a flowchart to be integrated with the revised methods section discussed above, to more clearly highlight the pipeline of our method.
>
> > In the motivation, the authors mentioned the importance of rotational and translational invariance. Therefore, it would be relevant to discuss methods like SE3-equivariant approaches that handle these invariances.
>
> We will discuss them in the introduction as well as in the methods section, reflecting the inclusion of equivariant models in the experiments.
>
> > The paper lacks in-depth discussions on limitations, such as issues regarding model robustness and ethical considerations, particularly regarding data misuse.
>
> We will include a more in-depth discussion of limitations in the revised version. In particular, with respect to model robustness we note that MEKRR transfers much more effectively compared to other algorithms (see Table 2), but that it is integral to stress test the final model to confirm that it is up to task before deploying it to production. For ethical considerations and data misuse we are unsure what the reviewer means, in general MEKRR inherits the same ethical problems as the field at large.

---

> > ### Comment · Reviewer_7Ada · 2023-08-16
> >
> > Thank you for the authors' responses. I have decided to maintain my current rating, and hope that the authors will incorporate the mentioned changes in the revision.

---

### Author Rebuttal · Authors · 2023-08-09

We thank the reviewers for their useful comments, which we will incorporate in our revision. Before addressing each review in detail, in this global message we answer some key suggestions/questions from the reviewers that we implemented and summarize our contributions.

Note that we have updated Table 1 and Table 2 with results using SCN which may be found in the attached pdf file.

__Comparison to more recent baselines__

While the original experiments only focused on SchNet as a pretrained feature map, we have conducted additional experiments using SCN, which is a state-of-the-art architecture on the OC20 S2EF task and an equivariant GNN, as suggested by several reviewers. In particular, we used the feature map extracted from SCN to build MEKRR, both for the standard and the multi-species $\alpha$ version, and added the baselines of training SCN from scratch or fine-tuning the feature map on the new dataset.

We report the results of these additional experiments in the updated Table 1 and Table 2 (see attached pdf). From these results, we can draw the following conclusions:
* We confirm the original conclusions with MEKRR outperforming training the GNN from scratch or standard fine-tuning, while being orders of magnitudes faster to train.
* Using MEKRR with the pre-trained features obtained with SCN as opposed to SchNet in general leads to a slight improvement in performance.
* Tuning the multi-species $\alpha$ parameter leads, again, to further improvements.

These results support the generality of our approach.

We want to highlight the fact that we could not properly fine tune SCN (it should probably be run for more epochs, we run it for 200 epochs for Table 1) due to the time constraints. Below we show this explicitly by comparing the time needed to train MEKRR-SCN vs time to train SCN-FT for one epoch both in seconds on datasets $D_1$ to $ D_4$. Both methods use 8 as the representation layer.


| Algorithm                    | $D_1$ | $D_2$ | $D_3$ | $D_4$ |
| ---------------------------- | ----- | ----- | ----- | ----- |
| SCN-FT (seconds / epoch)     | 50    | 88    | 43    | 61    |
| MEKRR-SCN (seconds to train) | 60    | 119   | 53    | 85    |

Note how MEKRR-SCN __fully trains__ in the time it takes to train SCN-FT for __one epoch__.

__Our contributions:__

Several of the reviewers asked for clarification about the contributions and key points of novelty of our approach. Here we briefly summarize them and we will include an expanded discussion in the manuscript.

Our proposed method, MEKRR, confronts the open problem of learning interatomic potentials in settings where high quality data samples are scarce and/or extremely costly. The _sample complexity_ of an algorithm  — that is, the number of data points necessary to guarantee a small generalization error — is heavily influenced by the representation of the data, and MEKRR is designed around this fact.

Our construction combines a feature map pre-trained on a large dataset (OC20) with a fine-tuning scheme based on kernel mean embeddings. In this way we aim at extracting the best out of the representation power of GNNs and kernel methods for interatomic potential learning.

To the best of our knowledge, chaining kernel mean embeddings to deep-learned feature maps is new in this context, and we precisely choose kernel mean embeddings to satisfy the permutational symmetry inherent in atomistic datasets. GNN features, on the other hand, automatically takes care of roto-translational invariance/equivariance, making our method a natural fit for the problem of interatomic potential learning.

We point out that our approach shows __excellent out-of-distribution performance__. Indeed, the OC20 data points, on which the feature maps were pre-trained, are __definitely not__ sampled from physically relevant distributions (see e.g. section 5.2 of the draft). Despite this, MEKRR still outperforms the other baselines on the realistic datasets for the two specific systems Cu/Formate and Fe/N$_2$.

---

### Decision · Program_Chairs · 2023-09-21

**Decision:**

Accept (poster)

**Comment:**

This paper proposes a transfer learning algorithm for modeling the energy surface of atomic systems, where the representations obtained from pre-trained GNNs are combined with kernel mean embeddings. The paper demonstrated the generalizability and transferability of the proposed method, offering significant gains over baselines. Most of the reviewers are happy with the quality and significance of the present work.